# Vibrational Study on the Structure, Bioactivity, and Silver Adsorption of Silk Fibroin Fibers Grafted with Methacrylonitrile

**DOI:** 10.3390/molecules28062551

**Published:** 2023-03-10

**Authors:** Michele Di Foggia, Masuhiro Tsukada, Paola Taddei

**Affiliations:** 1Department of Biomedical and Neuromotor Sciences, University of Bologna, Via Irnerio 48, 40126 Bologna, Italy; 2Division of Applied Biology, Faculty of Textile Science and Technology, Shinshu University, 3-15-1 Tokida, Ueda 386-8567, Japan

**Keywords:** silk fibroin, *Bombyx mori*, grafting, methacrylonitrile, Raman spectroscopy, IR spectroscopy

## Abstract

Natural fibers have received increasing attention as starting materials for innovative applications in many research fields, from biomedicine to engineering. *Bombyx mori* silk fibroin has become a material of choice in the development of many biomedical devices. Grafting represents a good strategy to improve the material properties according to the desired function. In the present study, *Bombyx mori* silk fibroin fibers were grafted with methacrylonitrile (MAN) with different weight gains. The potential interest in biomedical applications of MAN functionalization relies on the presence of the nitrile group, which is an acceptor of H bonds and can bind metals. IR and Raman spectroscopy were used to characterize the grafted samples and the possible structural changes induced by grafting. Afterward, the same techniques were used to study the bioactivity (i.e., the calcium phosphate nucleation ability) of MAN-grafted silk fibroins after ageing in simulated body fluid (SBF) for possible application in bone tissue engineering, and their interaction with Ag^+^ ions, for the development of biomaterials with enhanced anti-microbial properties. MAN was found to efficiently polymerize on silk fibroin through polar amino acids (i.e., serine and tryptophan), inducing an enrichment in silk fibroin-ordered domains. IR spectroscopy allowed us to detect the nucleation of a thin calcium phosphate layer and the uptake of Ag^+^ ions through the nitrile group, which may foster the application of these grafted materials in biomedical applications.

## 1. Introduction

The functionalization of natural fibers has gained considerable interest among material scientists to create high-performing and environmentally friendly materials for traditional and innovative applications [1]. Natural fibers can be divided into two main groups according to their chemical composition: plant fibers based on polysaccharides (such as cotton or linen) and animal fibers based on proteins (i.e., silk or wool). Wool keratins and silk fibroins are biopolymers with outstanding properties that allow their application over a wide variety of fields, from biomedicine (as sutures [2], nanocarriers [3], hydrogels [4] or scaffolds [5]), to engineering (as electrical nanogenerators [6] or wearable electronic devices [7]). In recent years, silk fibroin has become a material of choice in the development of many biomedical devices [8,9,10].

Silk is produced by different species of insects (silkworms, such as *B. mori*) and spiders and is composed of proteins rich in glycine, serine, and alanine residues organized into α-helix or β-sheet nanocrystals embedded into a semi-amorphous matrix [11]. In particular, silk from *B. mori* is formed by two different proteins: fibroin, a semi-crystalline biopolymer, and sericin, a glue-like protein, which is usually removed by different degumming processes. While the former protein has shown biocompatibility comparable to commonly used biomaterials, the latter protein is responsible for adverse biological responses [12].

Organic or inorganic molecules have been used to functionalize silk fibroin with enhanced properties by cross-linking or grafting [13,14,15]. The graft copolymerization of vinyl monomers onto silk fibers started in the early 1960s in Japan and has been widely applied in the textile industry as an alternative to the traditional mineral weighting technique, used for decades to compensate for the mass loss resulting from degumming. In more recent years, grafting has been considered not only an effective method to increase silk weight, but also a good strategy to improve the material properties according to the desired function. Monomers such as methacrylamide (MAA), 2-hydroxyethyl methacrylate (HEMA), methyl methacrylate (MMA), and 4-hydroxybutyl acrylate (HBA) have been found effective in improving the functional performances of silk without altering its intrinsic properties [16,17,18,19,20,21,22].

In order to produce grafted silk fibers with improved properties, the study of their molecular conformation is essential. In this context, our investigation aimed to gain more insights into the structural modifications occurring in silk fibroin fibers upon grafting with methacrylonitrile (MAN) with different weight gains (ranging between 10% and 60%). Previous studies demonstrated that MAN did not alter silk fibroin crystalline structure, surface smoothness, and tensile properties [23], while increasing yarn size and reducing moisture absorption. Interestingly, MAN-treated fibers had higher alkaline stability and an increased decomposition temperature (i.e., increased thermal stability). The variation of some optical (i.e., birefringence) and thermal parameters (i.e., glass transition temperature) suggested that grafting occurred only on amorphous fibroin domains [24].

The study aimed to gain insight into the chemical mechanism of MAN grafting, leading to the above-mentioned physical changes. For this purpose, vibrational Raman and IR spectroscopy have proved a powerful tool for studying polypeptides and proteins’ secondary and tertiary structures and their conformational rearrangements.

Carbon materials containing significant quantities of nitrile groups have been reported as particularly attractive owing to the C≡N metal-binding ability, which can improve the materials’ performance in applications including metal capture [25]. In this light and in view of the possible application of these grafted silk fibroins as biomaterials with enhanced anti-microbial properties, their silver adsorption ability was evaluated. In fact, silver-loaded silk membranes have already been described as anti-microbial wound dressing [26,27,28,29]. Moreover, the bioactivity of the grafting fibers (i.e., their calcium phosphate nucleation ability) was evaluated in vitro by ageing studies in simulated body fluid (SBF), mimicking the composition of human plasma [5,30].

## 2. Results and Discussion

### 2.1. Untreated Silk Fibroin Fibers

#### 2.1.1. Raman Spectra

The Raman spectra of the untreated *B. mori* silk fibroin fibers (Figure 1 and Appendix A) showed the Amide I band at 1666 cm^−1^, the Amide III band at 1229 cm^−1^, and a weaker component at 1266 cm^−1^ (unordered conformations), compatible with the general β-sheet structure conformation [20]. Bands at 1163, 1085, 977, and 882 cm^−1^ confirmed this observation. Additionally, Raman spectra gave information about amino acids: the I_644_/I_622_ intensity ratio, multiplied by the factor of 1.25, allows us to estimate the ratio between the contents of Tyr and Phe residues. The untreated silk fibers had a ratio of 7.1, in good agreement with previous results [22].

#### 2.1.2. IR Spectra

The IR spectra in both the parallel and the perpendicular orientations (Figure 2, Figure 3 and Appendix A) substantially confirmed the same conformational organization: Amide I showed a shoulder around 1697 cm^−1^ (anti-parallel β-sheet conformation) and a strong band centered at 1616 cm^−1^ (parallel mode) and 1624 cm^−1^ (perpendicular mode). Amide II and III fell at 1508 cm^−1^ (perpendicular) and 1515 cm^−1^ (parallel), and 1260 and 1230 cm^−1^, respectively. All these spectral features revealed that silk fibroin had an overall β-sheet conformation [22,31]. The bands at 1000–976 cm^−1^ (parallel) and 996–975 cm^−1^ (perpendicular) are typical of Ala-Gly sequences in fibroin, which generate a close network of hydrogen bonds among fibroin chains, giving rise to the typical β-sheet crystalline structure of silk [20].

### 2.2. Methacrylonitrile-Grafted Silk Fibroin Fibers

#### 2.2.1. Raman Spectra

Upon grafting, Raman bands at 2989–2939–2873 cm^−1^ (CH_3_ and CH_2_ stretching [32,33,34,35]), 2237 cm^−1^ (C≡N stretching 32–35]), 1454–1403 cm^−1^ (antisymmetric CH_3_ bending and symmetric CH_2_ bending [32,33,34,35]), 1174 cm^−1^ (CC stretching + CH_3_ and CH_2_ bending [33]), 977 cm^−1^ (CC stretching + CH_3_ and CH_2_ bending [33]), 703–685 cm^−1^ (CH_3_ and CH_2_ out-of-plane deformation [33], C-CN stretching [32]), and 586 cm^−1^ (CH_3_ and CH_2_ rocking [33], CN bending [32]), strengthened due to MAN incorporation (Figure 1 and Appendix A). The absence of any strengthening at about 1625 cm^−1^, where C=C stretching falls [32,34], suggests the actual polymerization of MAN onto silk fibroin.

The Raman I_2237_/I_Amide I_ and I_703_/I_Amide I_ intensity ratios were the most suitable for spectroscopically monitoring grafting: the first band of each ratio is diagnostic of the grafting agent, while the Amide I band was chosen as an internal standard for silk, since it is not affected by the unreacted monomer (Figure 4). An almost perfect linear correlation was established between both intensity ratios and weight gain (I_2237_/I_Amide I_: R^2^ = 0.999; I_703_/I_Amide I_: R^2^ = 0.992). On the other hand, Wallasch et al. have used the Raman CN stretching vibration at about 2235 cm^−1^ to quantify MAN [36].

The position of the Raman CN stretching band could give insights into its H-bond state; the nitrile group can be the acceptor of hydrogen bonds (i.e., from water). In our spectra, the CN stretching mode was observed at 2237 cm^−1^. On the basis of this wavenumber position, we can exclude any possible hydrogen bond interaction with silk fibroin, as already observed in (acrylonitrile-co-methyl acrylate)-silk blends [37]. On the other hand, in the association between liquid acrylonitrile and Ni^2+^ ion [38], similar wavenumber values were reported for “free” CN groups (around 2245 cm^−1^).

Similarly, Raman spectra excluded the possibility that nitrile could be transformed into imino-ether (R-C(-OR’)=NH), forming a covalent bond with protein amino groups. This effect was previously observed in the immobilization of antibodies on electrografted polyMAN films, but only in basic conditions (pH > 9) [39], which differed from the acidic environment used in the grafting procedure (see Paragraph 3.1). Therefore, on the basis of our data, we may exclude the possibility that the nitrile group could be involved directly in the grafting mechanism.

The position and full width at half-maximum (FWHM) of the Amide I Raman band were poorly affected by grafting with MAN; FWHM differences were < 1 cm^−1^, and therefore, they were not considered significant. The trend of the I_1266_/I_1229_ intensity ratio (i.e., between the bands of the Amide III spectral region sensitive to unordered and β-sheet conformations, respectively) showed a slight decrease from 0.44 ± 0.02 in the untreated silk to 0.36 ± 0.02 in the grafted fibers with weight gain of 60%, suggesting a moderate decrease in the unordered conformation content. The graph in Figure 5 shows that the I_1266_/I_1229_ intensity ratio is correlated with weight gain, suggesting that this effect progressively increased with the amount of grafted MAN.

This trend was opposite to what was observed in HEMA and HBA-grafted *B. mori* fabrics [20] and MAA-grafted Tussah silk fibroin [19]: in the latter case, a rearrangement in the hydrogen bonds pattern of fibroin in crystalline domains due to grafting was proposed. Tsukada et al. previously observed [23] that the glass transition temperature of MAN-grafted fibers decreased since hydrogen bond rearrangement occurred only in the amorphous regions, further supporting our spectroscopic data.

In order to gain more insight into the grafting mechanism, several Raman intensity ratios relative to the diagnostic bands of polar amino acids were investigated. The present study used experimental conditions that proved suitable to maximize the grafting yield: an initiator forms free radicals that may react with carboxyl, amine, and hydroxyl side groups of different amino acids. This process leads to forming a fibroin macroradical that reacts with the monomer, leading to the propagation of a grafted polymer chain. In particular, hydroxyl and amine groups in the side chains of serine, threonine, and tryptophan amino acids have been reported to interact with electron donor groups such as sulfate [31]. Previous studies identified the OH group of Ser and the NH group of glycine as the most accessible grafting sites of fibroin [20]. Guan et al. investigated the grafting of a vinyl phosphate (diethyl-2-(methacryloyloxyethyl) phosphate) to silk fibroin: the quantitative analysis of amino acids before and after grafting showed that cysteine, serine, and tyrosine reacted with potassium persulfate (the initiator) creating the RCH_2_^●^ and RCH_2_S^●^ radicals, i.e., the reaction sites for grafting [40]. Pourjavadi et al. reported the involvement of secondary alcohols in grafting MAN on chitosan, using ammonium persulfate as an initiator, which can extract H from those groups [41].

The involvement of serine residues in the grafting mechanism of silk fibroins with acrylate monomers has been extensively investigated [19,21,22,31] using the intensity of the 1404 cm^−1^ band (bending vibration of serine hydroxyl group), which reduced upon grafting. In MAN-grafted samples, the I_1404_/I_Amide I_ Raman intensity ratio increased (in the range of 0.26–0.31) compared to the untreated sample (I_1404_/I_Amide I_ = 0.20 ± 0.04), as expectable based on the contribution of the CH_2_ bending vibration of grafted MAN (1405 cm^−1^, Figure 1). However, this ratio does not linearly increase with the MAN content (Appendix A); we interpreted this behavior by hypothesizing that simultaneously, the contribution of the δOH mode of serine decreased due to the involvement of this amino acid in grafting.

The 1555 cm^−1^ band is attributed to the indole ring vibration of tryptophan: the linear decrease of the I_1555_/I_Amide I_ Raman intensity ratio with MAN content may confirm the involvement of Trp residues in grafting (Appendix A).

The Raman tyrosine doublet at about 850–830 cm^−1^ has been widely used to describe the average hydrogen-bonding state of the tyrosine phenoxyl groups (and therefore, whether the tyrosine residues are buried or exposed) in globular proteins. This intensity ratio reduced linearly (Appendix A) from 1.44 ± 0.02 in the control fibers to 1.27 ± 0.07 in the MAN 60% grafted sample. This finding indicates a lower exposure of Tyr residues (i.e., an increased H-bond donor role of the phenolic group), as observed in silk fibroin grafted with acrylates and treated with aqueous methanol, in which this effect was linked to fibroin recrystallization into β-sheet (i.e., a more ordered structure) [20].

#### 2.2.2. IR Spectra

Figure 2 and Figure 3 show the IR spectra of silk fibroin fibers before and after MAN grafting at the highest weight gain (i.e., 60%) recorded in the parallel and perpendicular orientations, respectively. Appendix A, show the IR spectra of all the samples under study, recorded in the parallel and perpendicular orientations, respectively.

In the IR spectra, bands at about 2980 cm^−1^ (CH_3_ and CH_2_ stretching [33,34,35]), 2235 cm^−1^ (C≡N stretching [33,34,35]), 1470–1445 cm^−1^ (antisymmetric CH_3_ bending [33,34,35]), 1390 cm^−1^ (symmetric CH_3_ bending [32,33,34,35]), 1217 cm^−1^ (backbone skeletal CC stretching + CH_3_ and CH_2_ bending [33,35]), and 976 cm^−1^ (ν CC stretching and + vinyl CH out-of-plane bending [33,35]) were identified as markers of MAN grafting (Figure 2 and Figure 3). At increasing weight gain, these bands were observed with progressively increasing intensity (Appendix A); the A_1217_/A_1230_ absorbance ratio was found to correlate well with the MAN content (Figure 6).

The above-mentioned bands are typical of the polymer, while the strongest band of the monomer at about 940 cm^−1^ was not detected. The absence or shallow content of unreacted monomers was also described by Pourjavadi et al. about the grafting of MAN on chitosan, using ammonium persulfate as an initiator [41].

The IR spectra of the grafted samples confirmed the overall β-sheet structure by the presence of the 1697 cm^−1^ band (anti-parallel β-sheet) in all grafted samples, most evident in the perpendicular orientation (Figure 3), as well as by the components at 996 and 975 cm^−1^, identified as marker bands of Ala-Gly sequences of silk fibroin [42].

IR spectra allowed us to gain more insights into the structural changes caused by grafting and get information on the orientation degree of the fibers. The last parameter could be qualitatively evaluated by inspecting the IR spectra recorded by placing the fibers along one specific direction (parallel) and perpendicularly to it. Although the prevailing conformation remained β-sheet upon grafting, minor shifts were observed: in particular, all the Amide bands shifted towards higher wavenumbers. In the parallel orientation, going from the untreated fibers to the MAN 60% grafted sample, Amide A, Amide I, II, and III bands shifted from 3278, 1616, 1515, and 1260 cm^−1^ to 3282, 1621, 1516, and 1262 cm^−1^, respectively (Figure 2); analogous shifts were observed in the perpendicular orientation (Figure 3). All these shifts could be explained by considering that hydrogen bonds between adjacent silk chains weakened due to grafting, in agreement with what was observed with MAA grafting [19].

Upon grafting, the IR spectra evidenced structural rearrangements in *B. mori* silk fibroin towards a decreased disordered conformation, as can be seen from Appendix A, at increasing grafting, Amide I and II progressively narrowed, and Amide II appeared to weaken with respect to Amide I. The latter trend appeared more evident in the spectra recorded in the parallel orientation (Appendix A). Accordingly, Figure 7 and Figure 8 show that the FWHM of Amide I and II, the A_Amide I_/A_Amide II_ ratio (an index of structural regularity), and the Ω_Amide I/Amide II_ marker (orientational order parameter) correlated well with the MAN content in the fibers.

In the Amide III range, the A_1230_/A_1260_ was 1.66 in the silk fibroin untreated fibers and decreased to about 1.5 in all the grafted samples, with no significant differences between each other (Appendix A). Surprisingly, the grafting with only 10% MAN appeared to induce a change in the ratio and alter the silk fibroin conformation to some extent. According to Bhat and Nadiger [43], the trend of this ratio confirms that upon grafting, structural rearrangements towards a more ordered state occurred.

The Ω_Amide III_ orientational parameter showed no correlation with weight gain and no significant changes going from untreated silk fibroin fibers to MAN-grafted ones (Appendix A). This result was in agreement with previous studies on *B. mori* silk fibers grafted with MAN, which evidenced that the molecular orientation in the crystalline region of fibroin, investigated by the XRD technique, remains relatively unchanged after grafting [23,24]. At the same time, a previous study on similar grafted materials found a lower material’s birefringence, suggesting that most of the physical changes occurred in the amorphous regions, where vinyl monomers can more easily penetrate and bind to silk reactive sites [24]. On the other hand, neither MAA-grafted *B. mori* [22] nor Tussah [19] silk fibroin fibers showed any orientational rearrangements induced by grafting.

### 2.3. Silver Adsorption Tests

Appendix A shows the weight gain measured after ageing of the fibers in the Ag^+^-containing solution for 48 h. As can be easily seen, the sample containing the highest amount of MAN underwent the highest weight gain (3.8%) under these conditions; moreover, it must be noticed that also the silk fibroin control sample underwent a significant weight gain (2.8%). According to a previous study [29], the Ag^+^ uptake by silk fibroin may be explained by considering that the most probable binding sites in the protein backbone are the free carboxyl groups of aspartic and glutamic acid occurring in the amorphous polypeptide sequences. Since these acidic groups have a pKa of about 4–4.8, they are almost completely dissociated at pH 7.4, thus providing negatively charged groups available for binding metal cations [29].

Figure 9 shows that the % weight gain of the fibers due to Ag^+^ uptake linearly increased with the MAN content. In other words, the samples containing the highest amounts of MAN were able to adsorb higher amounts of silver.

Figure 10 and Figure 11 show the IR spectra of the 60% MAN-grafted fibers before and after ageing for 48 h in the Ag^+^ solution, recorded in the parallel and perpendicular orientations, respectively. Due to the brownish color of the aged fibers, the recorded Raman spectra were not of good quality and, thus, are not reported and discussed.

As can be easily seen, upon ageing, the CN stretching at 2235 cm^−1^ slightly shifted to a higher wavenumber value in both the parallel and perpendicular spectra (Figure 10 and Figure 11). No analogous changes were observed in the other grafted samples. A similar shift towards higher wavenumbers was reported for the interaction between polyMAN and lithium salts [44], as well as between nitriles and Ag^+^ and other metals [38,45]. Moreover, Zarembowitch and Maleki [32] have reported that upon Cu^+^ complexation by MAN, the CN stretching band shifted to higher wavenumbers. The frequency shift was explained by these authors by considering that copper (I) is bound to the nitrogen atom of the CN group and a strengthening of the σ C≡N system occurs; however, the existence of a relatively strong π back-bonding from copper(I) to nitrogen was revealed. In agreement with the literature, we may interpret our results as a sign of silver chelation.

In all the samples, the MAN content remained constant upon ageing, as revealed by the constancy of the A_1217_/A_1230_ absorbance ratio (Appendix A). In other words, no MAN release in the ageing medium was detected under the used experimental conditions.

In the parallel spectra of the 60% MAN-grafted fibers, the Amide I and II bands slightly shifted to higher wavenumbers (Figure 10) upon ageing; the former band also broadened, while such behavior was not detected for Amide II (Appendix A). EDTA-modified *B. mori* silk fibroin fibers showed a similar shift towards higher wavenumbers for both Amide I and Amide II bands after binding to Cu^2+^ and Co^2+^ ions [46]. In the perpendicular spectra, both Amide I and II bands broadened upon silver uptake (Figure 11 and Appendix A). The other grafted samples showed less significant changes, as expectable on the basis of their lower silver uptake (Appendix A); the silk fibroin control fibers showed a significant broadening of Amide I in the parallel spectra (Appendix A), according to their high silver uptake.

In the Amide III range, the A_1230_/A_1260_ absorbance ratio decreased in both parallel and perpendicular spectra of the 60% MAN-grafted sample upon ageing (Appendix A). The other grafted samples showed less significant changes, as expectable on the basis of their lower silver uptake; the control fibers showed a significant variation of the ratio in the perpendicular spectra, which generally showed more significant changes.

The graphs reported in Appendix A, showed that the broadening of Amide I (in the parallel spectra) and Amide II (in the perpendicular spectra) increased at increasing silver uptake. In particular, by excluding the 25% MAN-grafted sample (outlier) from the graphs, a very good linear trend was observed between the % variation in the FWHM of Amide I and Amide II and % weight gain (i.e., Ag uptake), with R^2^ values as high as 0.9499 (Appendix A).

Upon ageing, the A_Amide I_/A_Amide II_ ratio decreased in the parallel spectra of the 60% MAN-grafted fibers while increased in the perpendicular ones (Appendix A). Similar behavior was also observed for the other aged samples, particularly for the silk fibroin control fibers. The changes in the perpendicular spectra appeared more significant than those in the parallel ones, and a good linear trend was observed (R^2^ = 0.8866) between the % variation in the A_Amide I_/A_Amide II_ and A_1230_/A_1260_ ratios in perpendicular spectra and % weight gain (i.e., Ag uptake), by excluding the 25% MAN-grafted sample outlier (Appendix A). An analogously good correlation was found between Ω_Amide I/Amide II_ (Appendix A) and % weight gain due to Ag uptake (Appendix A). A similar A_Amide I_/A_Amide II_ trend was observed after the binding of Co^2+^ ions on EDTA and tannic acid-modified silk fibroins, while the binding of Cu^2+^ ions showed an opposite trend, suggesting that the metal–fiber interactions are strongly dependent on the kind of fiber and the metal cation [46].

The reported results evidenced Ag^+^ chelation and structural rearrangements in silk fibroin, which were found to correlate with silver uptake.

### 2.4. Bioactivity Tests in SBF

#### 2.4.1. Raman Spectra

After 7 days of immersion in the SBF solution, poor weight variations in all samples (either untreated or grafted) were measured, ranging from negative values (−1.5% in the control sample, corresponding to a weight loss) to slightly positive values (+1% in MAN 40% and MAN 60% samples), as reported in Appendix A. Based on these data, only these last two samples (with the control as a reference) were immersed in the SBF solution for up to 28 days to study their bioactivity at longer times. After 28 days in SBF, only the MAN 60% sample showed a moderate weight gain (Appendix A), while the control and MAN 40% showed a weight loss ranging from 4 to 6%. Based on the data reported in Appendix A, it appears that the presence of the grafted polymer protected the fibers against weight loss. In other words, the higher the MAN content, the lower the weight loss: the data obtained at 7 days support a linear correlation between weight gain after immersion in the SBF solution and MAN content % (Figure 12), while, due to the reduced number of samples, the correlation is less robust at 28 days of immersion.

Figure 13 shows the Raman spectra of silk fibroin control and grafted fibers (MAN 40% and MAN 60% samples) before and after immersion in SBF for 7 and 28 days.

Raman spectroscopy could not detect any nucleation of an inorganic phase even on the samples having positive weight gains, since the technique is more sensitive to the bulk of the material [30]. However, it proved suitable to determine whether the samples’ weight variation was associated with a mass loss due to the grafting agent. This was possible thanks to the previously mentioned Raman I_2237_/I_Amide I_ and I_703_/I_Amide I_ intensity ratios (Figure 4), which were found to be highly correlated with weight gain. Appendix A, report the trends of the I_2237_/I_Amide I_ and I_703_/I_Amide I_ intensity ratios before and after immersion of the fibers in the SBF solution for 7 and 28 days. The regression lines reported in Figure 4 allowed us to estimate the MAN content % of the grafted samples before and after the bioactivity tests in SBF (Table 1). Both intensity ratios revealed that none of the samples showed any significant variation in the MAN content % after 7 days of immersion in SBF; therefore, at this stage, no loss of the grafted polymer occurred. On the contrary, both ratios agreed about a significant loss of the grafting agent after 28 days of immersion in the SBF solution: both grafted samples showed a reduction of the above-mentioned intensity ratios by about 12–14% (Appendix A), which corresponded to an effective reduction of MAN content by about 5–7% (Table 1).

The Raman spectroscopic parameters, already discussed for the samples before the bioactivity tests, were poorly affected by immersion in the SBF solution (Figure 13). In particular, the position and the FWHM of the Amide I band did not change. The same result was observed after 7 days of immersion in SBF of silk fibroin grafted with phosphorylated methacrylates [30]. The trend of the I_1266_/I_1229_ intensity ratio (Appendix A) suggested the occurrence of little structural changes after 7 days of immersion. After 28 days of ageing, the I_1266_/I_1229_ intensity ratio increased by 50% for MAN 40%, and 15% for MAN 60%, suggesting an increase in structural disorder of the fibers, which was not detected in control fibers. This result could appear unexpected, since previous studies on the proteolytic degradation of silk fibroin fibers reported an increase in structural order (revealed by the narrowing of the Raman Amide I band) attributed to the preferential degradation of amorphous domains [47]. The opposite trend observed in the present study may be explained in relation to the loss of the grafted MAN polymer: as reported above, its introduction in the fibers had determined an increase in order (see Paragraph 2.2); thus, it is not surprising that when it is removed, the degree of disorder increased., no changes in the I_1266_/I_1229_ intensity ratio were detected in the control silk fibroin (i.e., no grafted) fibers.

Raman intensity ratios of the amino acids involved in the grafting reaction were more affected by the bioactivity tests: the I_1555_/I_Amide I_ ratio, sensitive to tryptophan (Appendix A), generally decreased even after 7 days of immersion. Opposite to what observed for the structural disorder, the control sample showed a higher decrease (−58% after 28 days) than the grafted samples (−40% and −28% after 28 days for MAN 40% and MAN 60% samples, respectively), further strengthening the idea that the grafted MAN polymer has a “protective role” against weight loss. The intensity decrease of Trp bands has been previously reported during the proteolytic degradation of Tussah silk fibers [47] and the alkaline hydrolysis of *B. mori* silk fibroin grafted with MAA [22]. This reduction was attributed to bulky Trp residues in the amorphous domains.

The I_1404_/I_Amide I_ intensity ratio (Appendix A), diagnostic for serine, showed a similar trend: an overall reduction after 7 days, followed by a more prominent reduction after 28 days. For this intensity ratio, the reduction was similar in control and grafted samples and ranged between 30 and 39%.

The I_850_/I_830_ intensity ratio (Appendix A), describing the H- bonding state of tyrosine, was the least affected by the immersion in SBF: after 28 days, control and grafted samples showed a minimal decrease by about 7–9%. A more marked decrease in this intensity ratio was observed during the proteolytic degradation of Tussah silk fibers [47]: the effect was explained by the cleavage of exposed Tyr residues.

The partial fiber degradation detected by Raman spectroscopy (with loss of MAN and amino acid residues) suggests that the weight gain results reported in Appendix A, represent the sum of the contributions from two opposite phenomena, i.e., mass loss due to fiber degradation and mass gain due to calcium phosphate nucleation. The latter process was investigated by IR spectroscopy.

#### 2.4.2. IR Spectra

IR spectra allowed us to gain more insights into the possible occurrence of calcium phosphate deposition, being sensitive to the surface of the samples (i.e., their first two microns). As an example, Figure 14 and Figure 15 show the IR spectra recorded in the parallel and perpendicular orientations on the 60% MAN-grafted fibers before and after ageing in SBF for 28 days.

In agreement with a previous study [30], the A_1060_/A_Amide I_ absorbance ratio was calculated to evaluate the nucleation of a calcium phosphate phase; in fact, near 1000 cm^−1^, the phosphate antisymmetric stretching has been reported to fall [48]. The values of this spectroscopic marker are reported in Figure 16.

As can be easily seen from the spectra reported in Figure 14 and Figure 15, the 1060 cm^−1^ band underwent a slight increase in relative intensity, and accordingly, the A_1060_/A_Amide I_ absorbance ratio increased as well (Figure 16); the changes appeared more significant in the perpendicular spectra and suggested the deposition of a thin calcium phosphate layer. The % variation in the A_1060_/A_Amide I_ absorbance ratio upon ageing in SBF for 28 days was found to correlate with the weight gain of the samples after this treatment (Appendix A). The formation of a calcium phosphate layer on silk fibroin can be explained by the interaction of calcium ions with the carboxylic residues of amino acids with a minor contribution of hydroxyl and carbonyl groups [48]. These types of interaction explained the extremely rapid mineralization upon immersion in the SBF solution of silk fibroin functionalized with anionic polypeptides, in which negative carboxylate ions acted as nucleation sites for the apatitic phase [5]. These results were of paramount importance in explaining the proliferation and adhesion of mesenchymal stem cells on the materials and, thus, their biocompatibility.

Upon ageing, the CN stretching at 2235 cm^−1^ did not show any significant change in its wavenumber value (Figure 14 and Figure 15), suggesting that the CN group did not chelate calcium ions.

At IR spectroscopy, the MAN content remained constant upon ageing for 28 days, as revealed by the constancy of the A_1217_/A_1230_ absorbance ratio (Appendix A). This result could appear in contrast with the Raman findings. In fact, it must be recalled that the two techniques have different sampling areas: Raman spectroscopy is sensitive to the sample bulk, while IR spectroscopy is sensitive to the sample surface. Therefore, IR spectroscopy revealed that no MAN release in the ageing medium occurred from the sample surface, while the MAN content of the sample bulk was affected by the ageing in SBF.

In agreement with the Raman findings, in the parallel spectra of the 60% MAN-grafted fibers (Figure 14), Amide I broadened upon ageing, while Amide II did not (Appendix A); an analogous behavior was already observed for the samples aged in the Ag^+^-containing solution (Paragraph 2.3). No changes were observed in the FWHM of Amide I and II bands in the perpendicular spectra (Figure 15).

In the parallel spectra of the 60% MAN-grafted fibers (Figure 14), the A_Amide I_/A_Amide II_ absorbance ratio slightly decreased (Appendix A), while it increased in the perpendicular ones (Figure 15 and Appendix A), as already observed for the samples aged in the Ag^+^-containing solution (Paragraph 2.3). Changes in this ratio were also observed in dentin collagen upon calcium loss or chelation (i.e., demineralization or remineralization) [49]. Accordingly, the Ω_Amide I/Amide II_ also had an analogous decrease (Figure Appendix A).

Additionally, the Amide III region underwent the same variations observed upon silver uptake, i.e., a general decrease of the A_1230_/A_1260_ absorbance ratio (Appendix A). As observed upon immersion in the Ag^+^-containing solution, the % variation in this ratio correlates well with weight gain upon ageing in SBF (Appendix A).

The observed trends allowed us to conclude that the chelation of silk fibroin with calcium and silver ions induced similar changes in the spectroscopic markers and, thus, in protein conformation. The presence of the grafted MAN polymer favored both processes. As a difference, the CN groups were found to participate in silver chelation, while no clear sign of this mechanism for calcium was inferred.

## 3. Materials and Methods

### 3.1. Materials

*B. mori* silk fibers were degummed before grafting and then treated with a mixture containing 2.5% sodium persulfate (the initiator), 2 mL/L formic acid (85%), 12% nonionic surfactant, and various amounts of methacrylonitrile, as previously described [23]. The mixture was heated at room temperature to 80 °C for 20 min and then maintained at the same temperature for 40 min. At the end of the reaction, the samples were washed with water. Fibers were extracted with a 1 g/L sodium hydrosulfate solution containing 1 mL/L nonionic surfactant (Noigen EC, Dai-ichi Kogyo Seiyaku Co., Tokyo, Japan) at 70 °C for 20 min to remove the unreacted MAN and washed. Silk fibers were dried in a forced draft at 100–105 °C for 2 h, placed in a desiccator over silica gel for 30 min, and weighed. The weight gain of silk fibers treated with MAN was calculated based on the oven-dried weights of the sample before and after the treatment: silk fibers with a weight gain of 10, 25, 32.5, 40, and 60% were prepared, the latter after 4 h of reaction time.

### 3.2. Silver Adsorption Tests

The adsorption and binding of silver cations onto untreated and grafted silk fibers were evaluated after immersing the samples in aqueous solutions of 0.5 mM AgNO_3_ and KNO_3_ at 37 °C for 48 h, as described elsewhere [29]. The solution pH of 7.4 was adjusted using ammonia.

The samples were then washed thoroughly with water, air-dried, and weighed in order to calculate the % weight gain according to the equation:% weight gain = 100 (w_48_ − w_0_)/w_0_
where w_48_ is the dry weight after 48 h of immersion in the Ag^+^-containing solutions and w_0_ is the initial weight.

### 3.3. Bioactivity Tests in SBF

The bioactivity of control and grafted silk fibroin fibers (test samples of 25 ± 5 mg) was assessed by immersing the samples into a simulated body fluid (SBF) buffered at pH 7.4 at 37 °C and containing the concentrations of the following ions: 142 mM Na^+^, 4 mM K^+^, 2.5 mM Ca^2+^, 148.8 mM Cl^−^, 4.2 mM HCO_3_^−^, 1 mM HPO_4_^−2^ [50]. The samples were then washed thoroughly with water, air-dried, and weighed in order to calculate the % weight gain according to the equation:% weight gain = 100 (w_t_ − w_0_)/w_0_
where w_t_ is the dry weight after 7 or 28 days of immersion in SBF and w_0_ is the initial weight.

### 3.4. Vibrational Spectroscopy

Raman and IR spectroscopy allowed the fibers’ characterization before and after grafting.

Raman spectra were recorded on a Bruker MultiRam FT-Raman spectrometer equipped with a cooled Ge-diode detector. The excitation source was an Nd^3+^-YAG laser (1064 nm) in the backscattering (180°) configuration, with a power of 100 mW. The focused laser beam diameter was about 100 μm, and the spectral resolution was 4 cm^−1^. The presented spectra are the average of three replicates: each replicate is the average of 10,000 measures. IR spectra were recorded on a Shimadzu IRTracer-100 FT-IR spectrometer, equipped with a QATR-10 single crystal diamond Attenuated Total Reflectance (ATR) accessory and a Deuterated Lanthanum α-Alanine doped TriGlycine Sulphate (DLaTGS) detector; the spectral resolution was 4 cm^−1^ with 64 scans for each spectrum. Fibers are intrinsically oriented samples; thus, their vibrational spectra depend on fiber orientation. Consequently, silk fibers were placed in the sample compartment keeping the same orientation for all the samples. The IR spectra were recorded in triplicate by positioning the fibers along one specific direction (conventionally termed “parallel”) and perpendicularly to it. Recording fibers’ spectra in different orientations allowed us to calculate some spectral parameters indicating orientational order, namely Ω_Amide III_ and Ω_Amide I/Amide II_; they were calculated as follows:Ω_Amide III_ = X_perp_/X_par_
Ω_Amide I/Amide II_ = R_par_/R_perp_
where X_perp_ and X_par_ were the A_1230_/A_1260_ absorbance ratios (as peak heights) calculated from the perpendicular and parallel spectra, respectively, and R_par_ and R_perp_ were the A_Amide I_/A_Amide II_ intensity ratios (as peak heights) calculated from the parallel and perpendicular spectra, respectively [21].

## 4. Conclusions

In this study, *B. mori* silk fibroin fibers grafted with methacrylonitrile (MAN) were characterized using Raman and IR spectroscopies to gain information on their structural modifications, bioactivity, and metal (Ag) adsorption.

The Raman technique allowed us to evaluate the content of the grafting agent, study the grafting mechanism, which involved Ser and Trp residues, and evaluate the minor structural changes induced by grafting (i.e., the decrease of unordered domains). The IR technique confirmed the previous findings and allowed us to study the effects of grafting on the orientational order/disorder of fibers.

The same techniques were applied to study the grafted samples’ metal binding and bioactivity. Silver adsorption was found to be proportional to the MAN content of fibers, caused a shift towards higher wavenumbers of the nitrile group, and induced minor structural changes on silk fibroins, evidenced by the shift and increased FWHM of the Amide I and Amide II bands. The bioactivity test in SBF showed similar structural variations compared to Ag^+^ adsorption, but evidenced a protective role of the grafting agent, which showed only a minimal release (5–7% in weight) after 28 days of immersion in the SBF solution and prevented silk fibroin weight loss. IR spectroscopy allowed the detection of a calcium phosphate phase on modified fibers, which was higher in the samples incorporating higher amounts of the grafting agent.

These preliminary results were encouraging about the positive effect of grafting on the bioactivity and metal adsorption properties of MAN-grafted silk fibroin fibers fostering their possible application as biomedical materials.

## Figures and Tables

**Figure 1 molecules-28-02551-f001:**
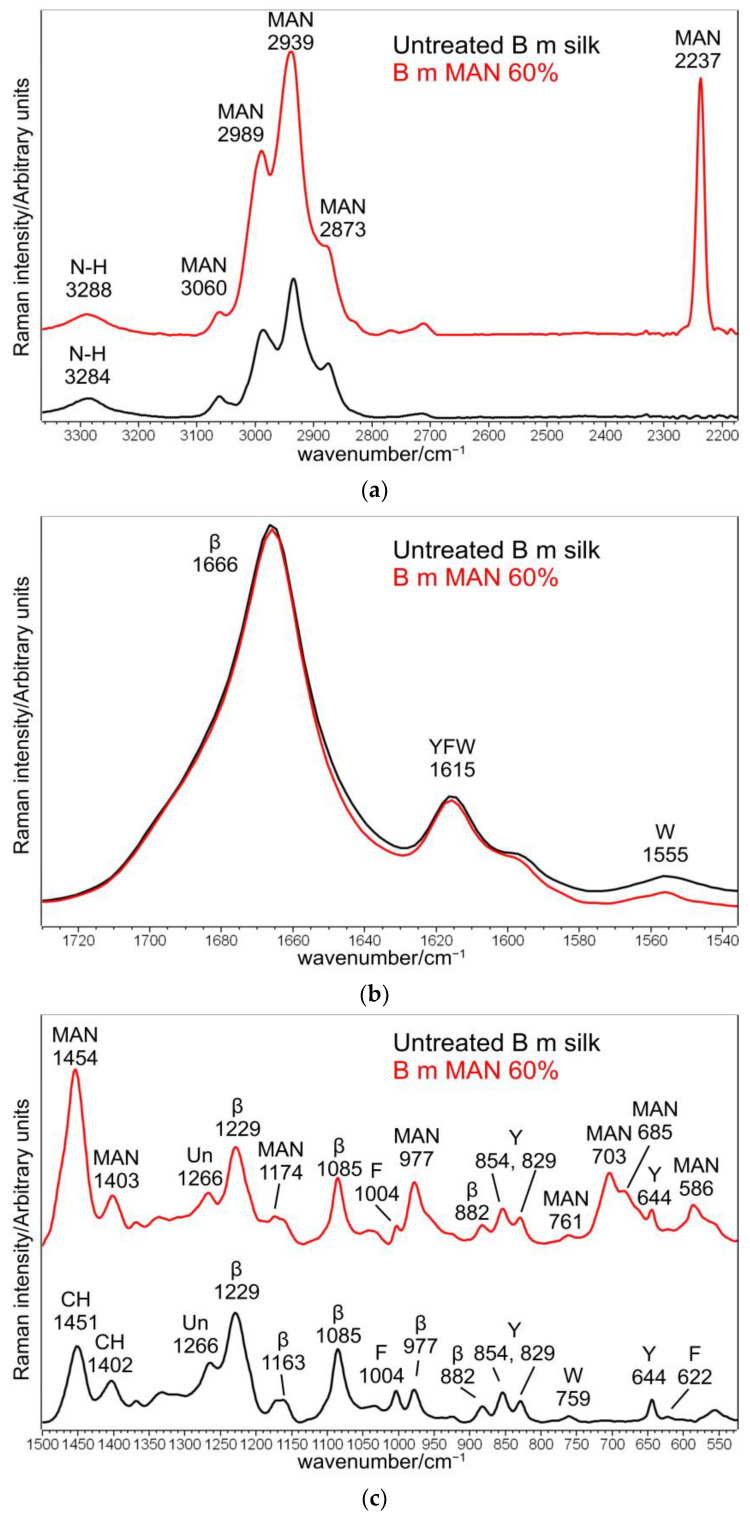
Raman spectra of *B. mori* silk fibroin fibers before and after grafting with MAN at the highest weight gain (i.e., 60%) in three different spectral ranges: (**a**) 3360–2180 cm^−1^, (**b**) 1730–1530 cm^−1^, and (**c**) 1500–530 cm^−1^. The spectra are normalized to the intensity of the Amide I band. The main bands assignable to β-sheet (β) or unordered (Un) conformation as well as to tyrosine (Y), phenylalanine (F), tryptophan (W), and the grafting agent (MAN) are indicated.

**Figure 2 molecules-28-02551-f002:**
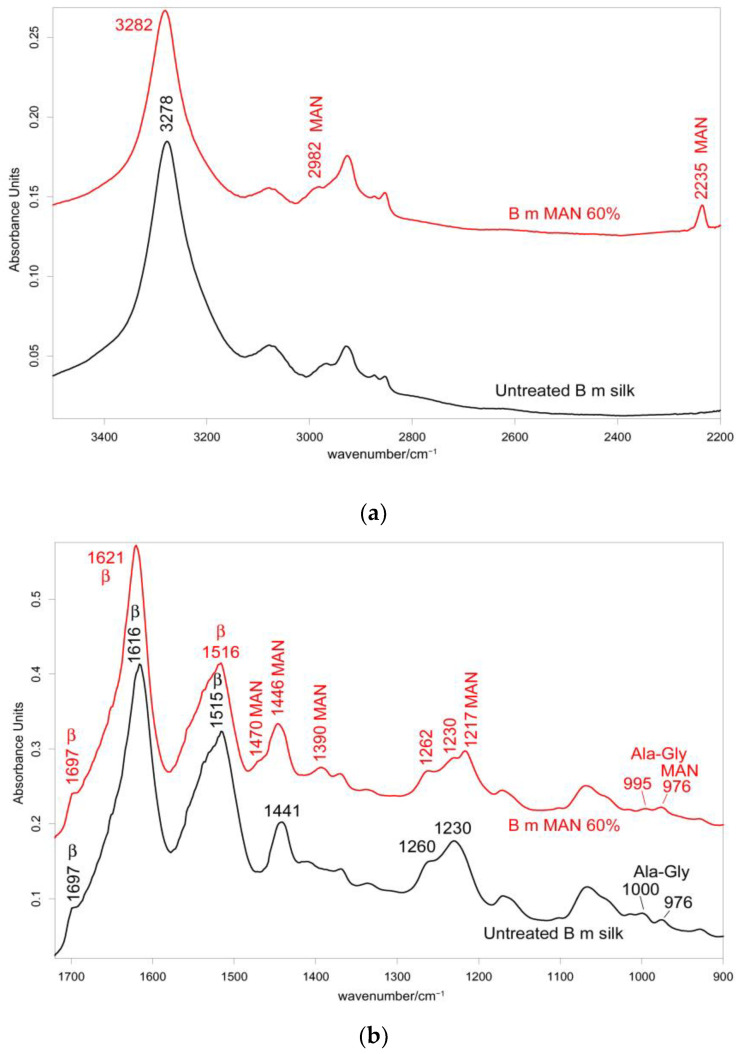
IR spectra recorded in the parallel orientation on *B. mori* silk fibroin fibers before and after grafting with MAN at the highest weight gain (i.e., 60%) in the 3500–2200 cm^−1^ (**a**) and 1720–900 cm^−1^ (**b**) spectral ranges. The spectra are normalized to the intensity of the Amide I band (1616–1621 cm^−1^). The main bands assigned to β-sheet (β) conformation, as well as to alanine-glycine domains (Ala-Gly), are indicated together with those having a contribution from MAN.

**Figure 3 molecules-28-02551-f003:**
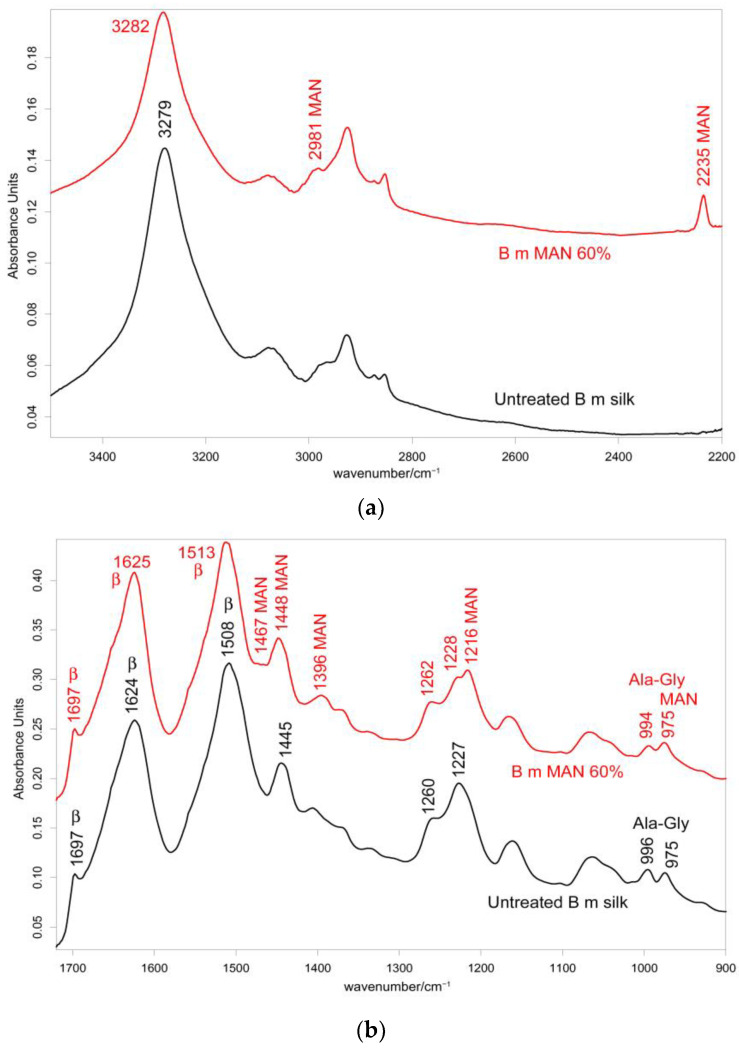
IR recorded in the perpendicular orientation on *B. mori* silk fibroin fibers before and after grafting with MAN at the highest weight gain (i.e., 60%) in the 3500–2200 cm^−1^ (**a**) and 1720–900 cm^−1^ (**b**) spectral ranges. The spectra are normalized to the intensity of the Amide I band (1624–1625 cm^−1^). The main bands assigned to β-sheet (β) conformation, as well as to alanine-glycine domains (Ala-Gly), are indicated together with those having a contribution from MAN.

**Figure 4 molecules-28-02551-f004:**
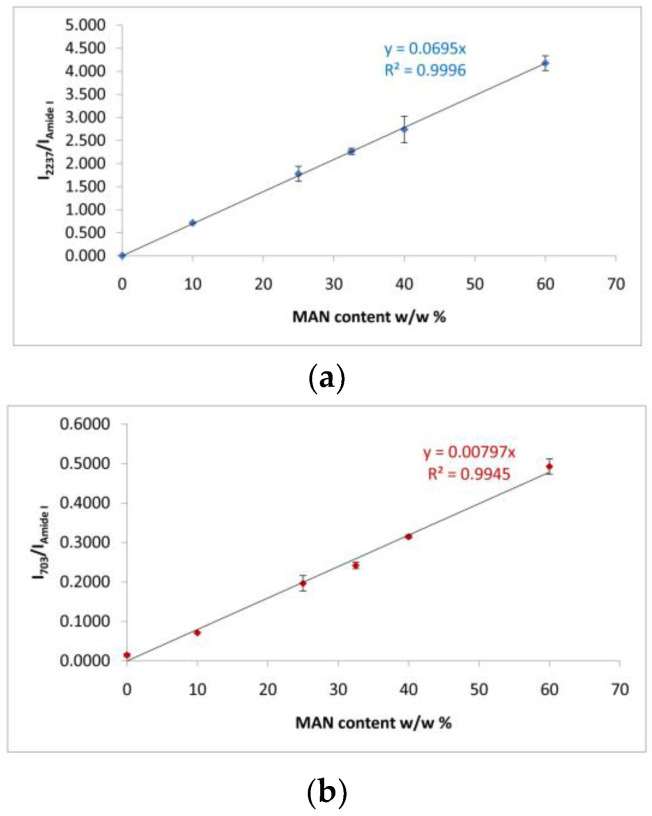
(**a**) Trend of the I_2237_/I_Amide I_ and (**b**) I_703_/I_Amide I_ Raman intensity ratios as a function of the weight gain (i.e., MAN content *w*/*w*%) for the silk fibroin fibers grafted with MAN.

**Figure 5 molecules-28-02551-f005:**
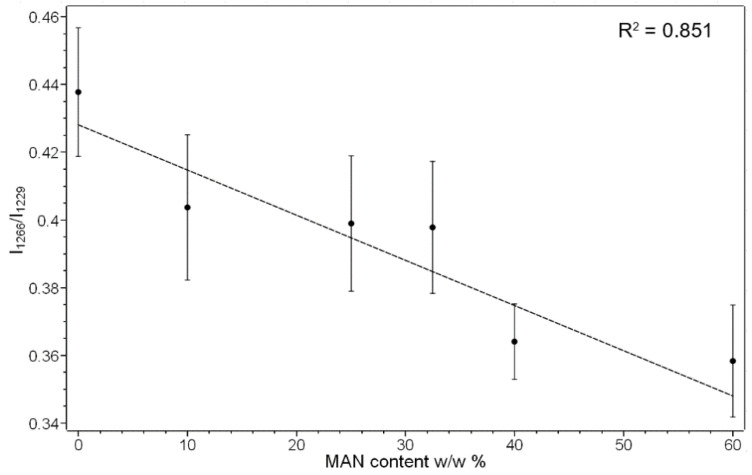
Trend of the I_1266_/I_1229_ Raman intensity ratio as a function of the weight gain (i.e., MAN content *w*/*w*%) for the silk fibroin fibers grafted with MAN.

**Figure 6 molecules-28-02551-f006:**
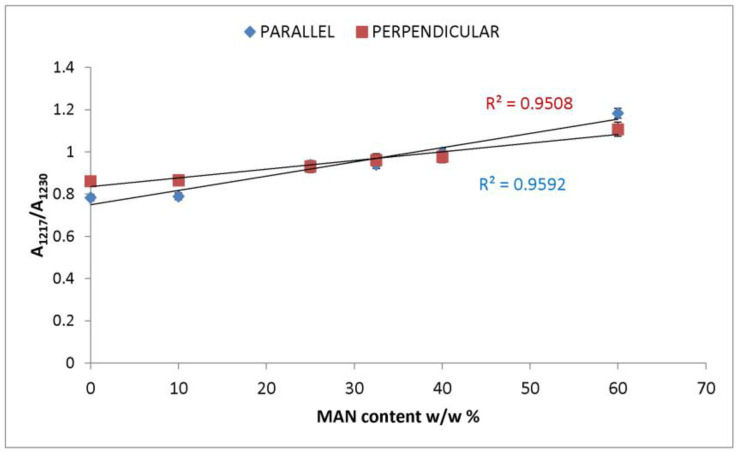
Trend of the IR A_1217_/A_1230_ absorbance ratio (obtained from the parallel and perpendicular spectra) as a function of the weight gain (i.e., MAN content *w*/*w*%).

**Figure 7 molecules-28-02551-f007:**
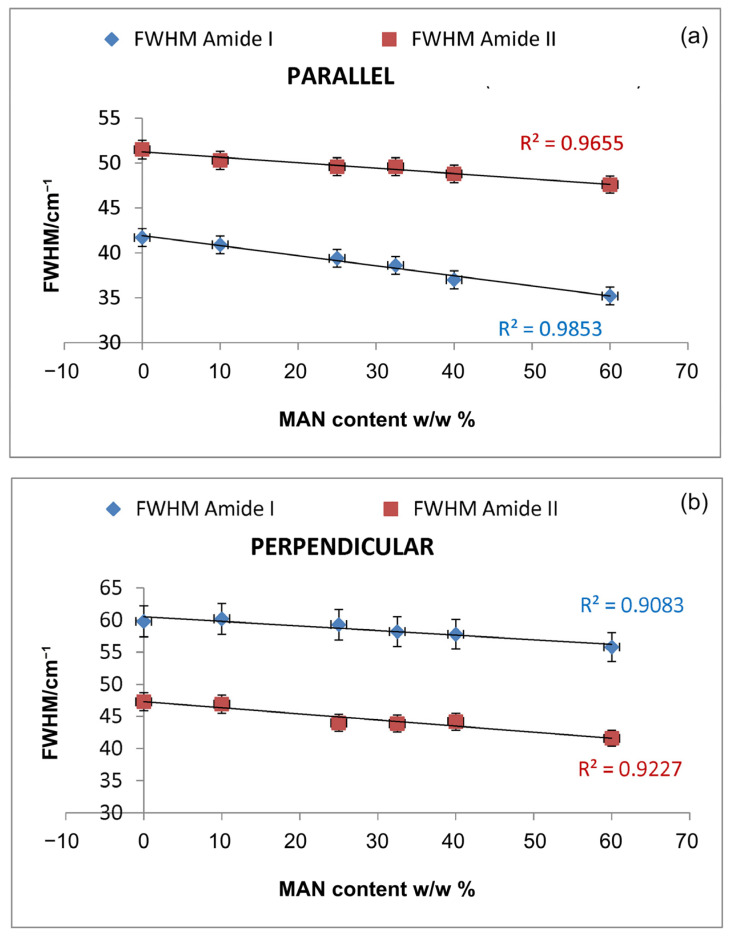
Trend of the FWHM of Amide I and II bands as a function of the weight gain (i.e., MAN content *w*/*w*%) in the parallel (**a**) and perpendicular (**b**) spectra.

**Figure 8 molecules-28-02551-f008:**
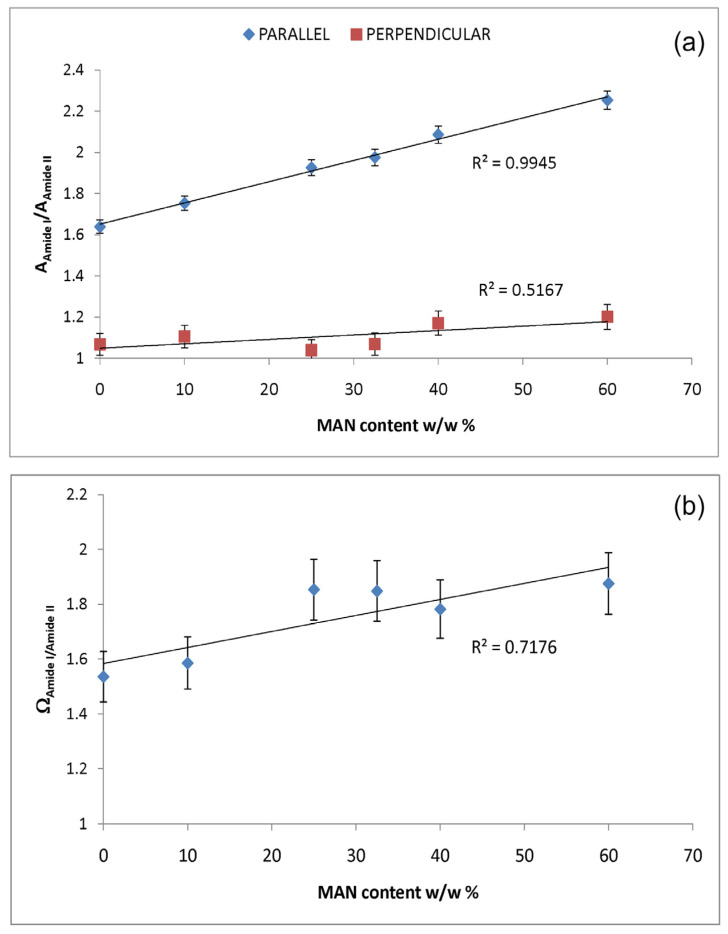
Trend of the IR A_Amide I_/A_Amide II_ absorbance ratio (**a**) in the parallel and perpendicular spectra and Ω_Amide I/Amide II_ marker (**b**) as a function of the MAN content (*w*/*w*%).

**Figure 9 molecules-28-02551-f009:**
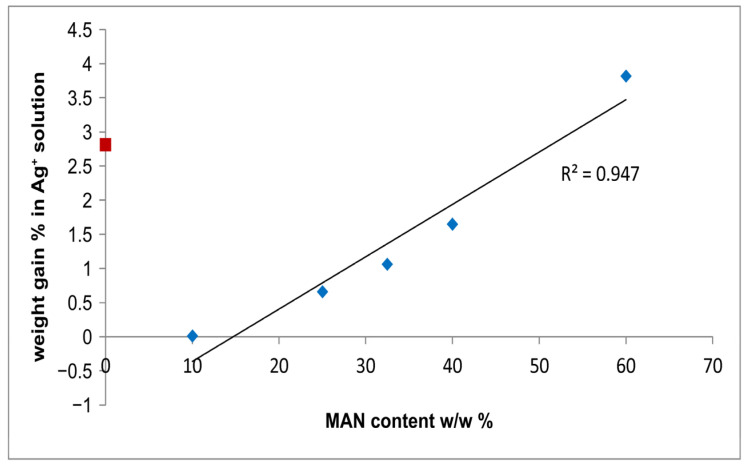
Trend of the % weight gain of the fibers due to Ag^+^ uptake as a function of the MAN content (*w*/*w*%). The silk fibroin control sample is indicated with a red square because it was not used to calculate the regression line.

**Figure 10 molecules-28-02551-f010:**
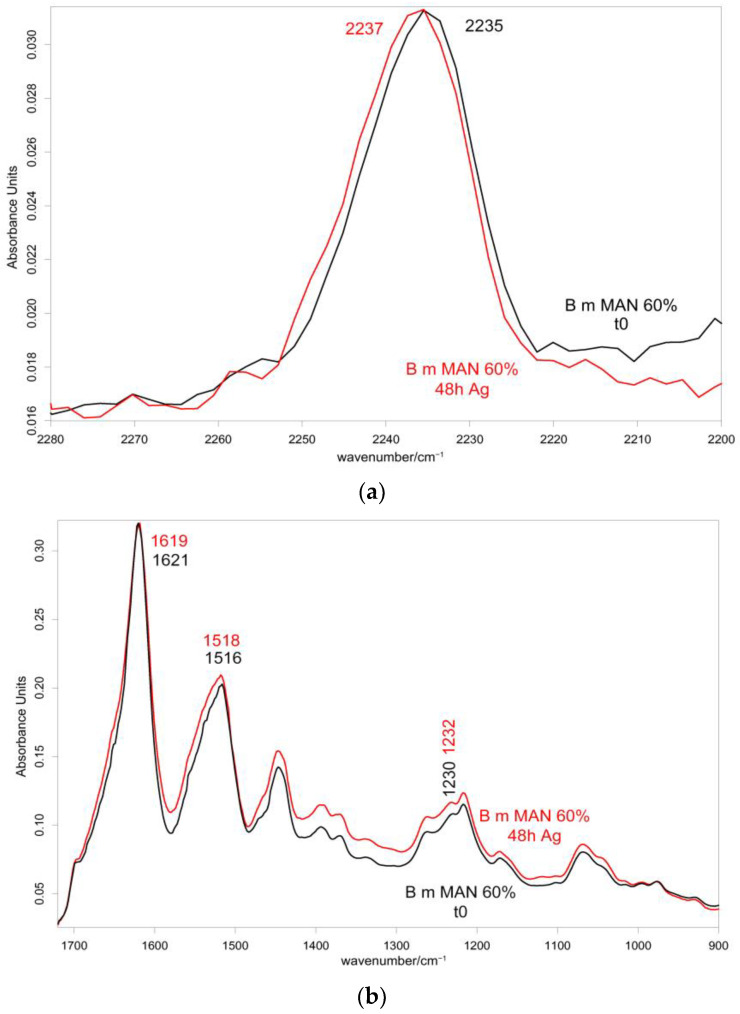
IR spectra recorded in the parallel orientation on the 60% MAN-grafted fibers before and after ageing for 48 h in the Ag^+^ solution in the 2280–2200 cm^−1^ (**a**) and 1720–900 cm^−1^ (**b**) spectral ranges.

**Figure 11 molecules-28-02551-f011:**
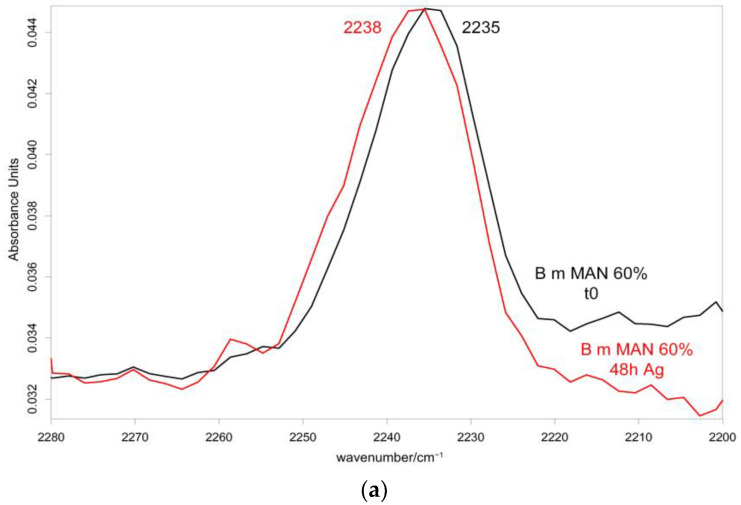
IR spectra recorded in the perpendicular orientation on the 60% MAN-grafted fibers before and after ageing for 48 h in the Ag^+^ solution in the 2280–2200 cm^−1^ (**a**) and 1720–900 cm^−1^ (**b**) spectral ranges.

**Figure 12 molecules-28-02551-f012:**
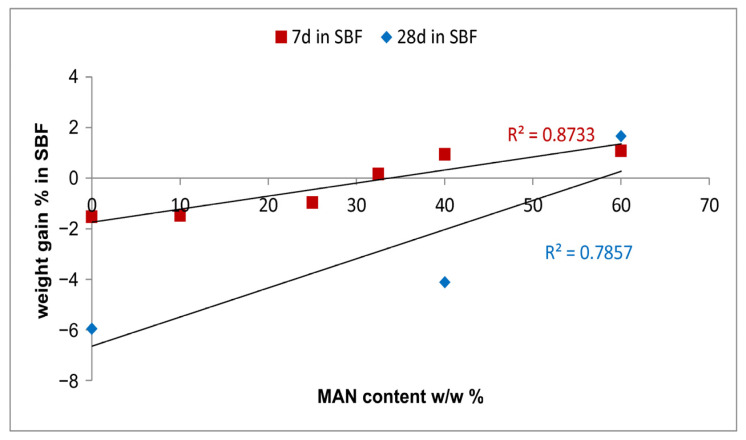
Trend of the % weight gain of the fibers upon ageing in SBF for 7 and 28 days as a function of the MAN content (*w*/*w*%).

**Figure 13 molecules-28-02551-f013:**
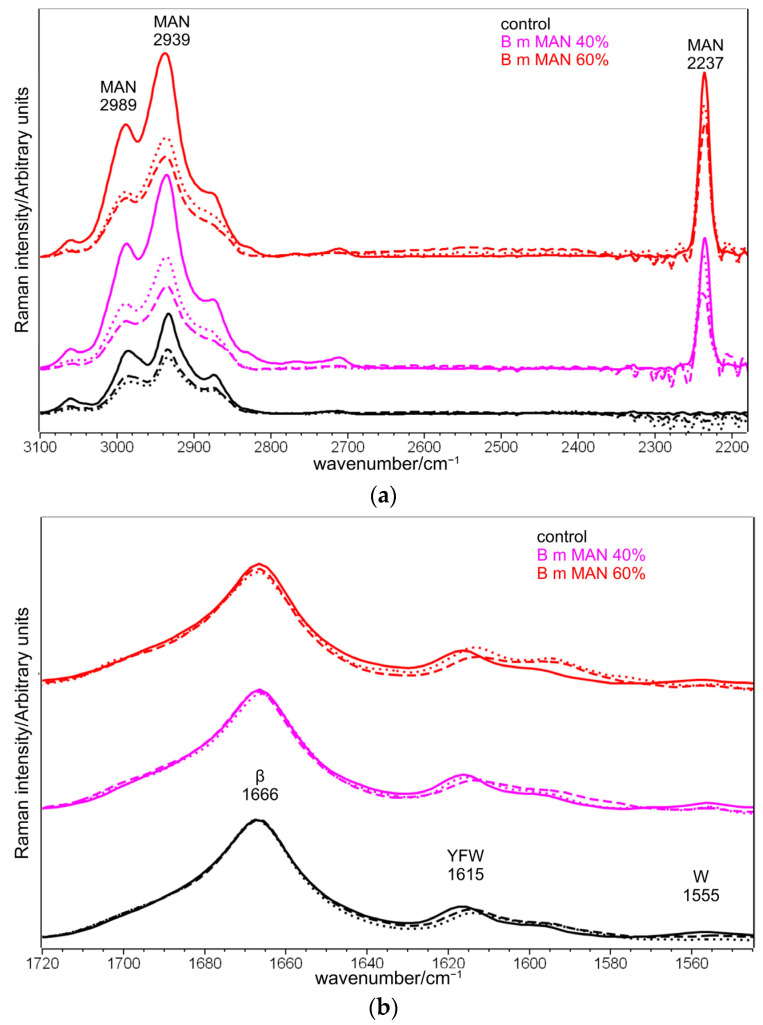
Raman spectra of *B. mori* silk fibroin control and grafted fibers (MAN 40% and MAN 60% samples only) before (continuous lines) and after immersion in SBF for 7 days (dotted lines) and 28 days (dashed lines) in three different spectral ranges: (**a**) 3100–2180 cm^−1^, (**b**) 1720–1540 cm^−1^, and (**c**) 1500–530 cm^−1^. The spectra are normalized to the intensity of the Amide I band. The main bands assignable to β-sheet (β) or unordered (Un) conformation as well as to tyrosine (Y), phenylalanine (F), tryptophan (W), serine (S), and the grafting agent (MAN) are indicated.

**Figure 14 molecules-28-02551-f014:**
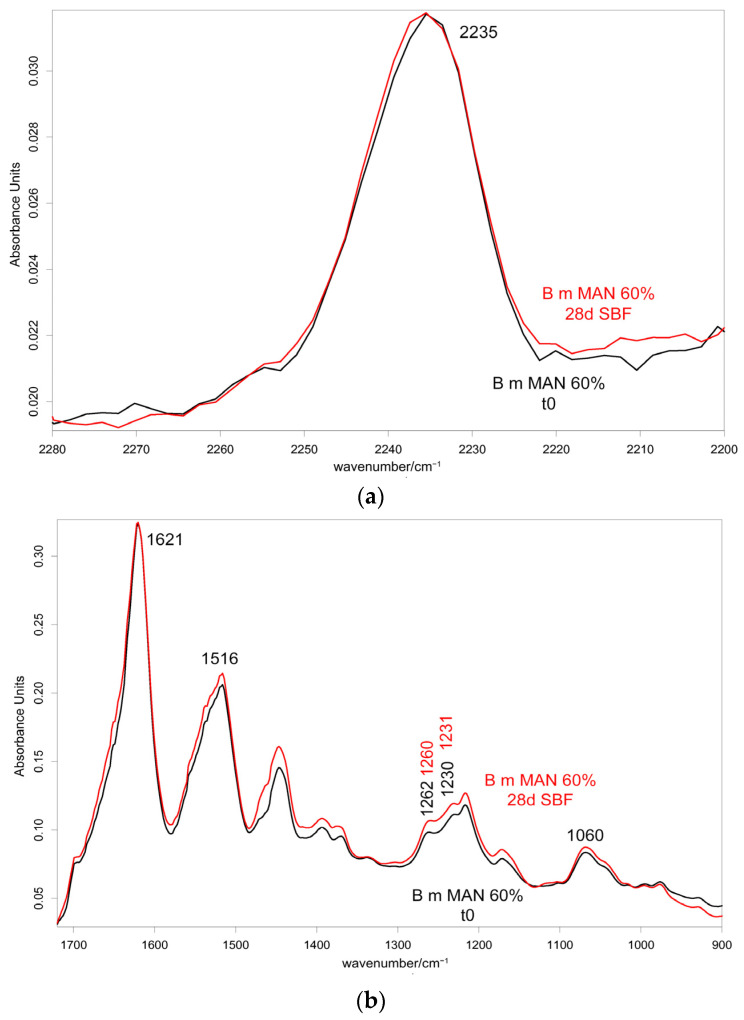
IR spectra recorded in the parallel orientation on the 60% MAN-grafted fibers before and after ageing for 28 days in SBF in the 2280–2200 cm^−1^ (**a**) and 1720–900 cm^−1^ (**b**) spectral ranges.

**Figure 15 molecules-28-02551-f015:**
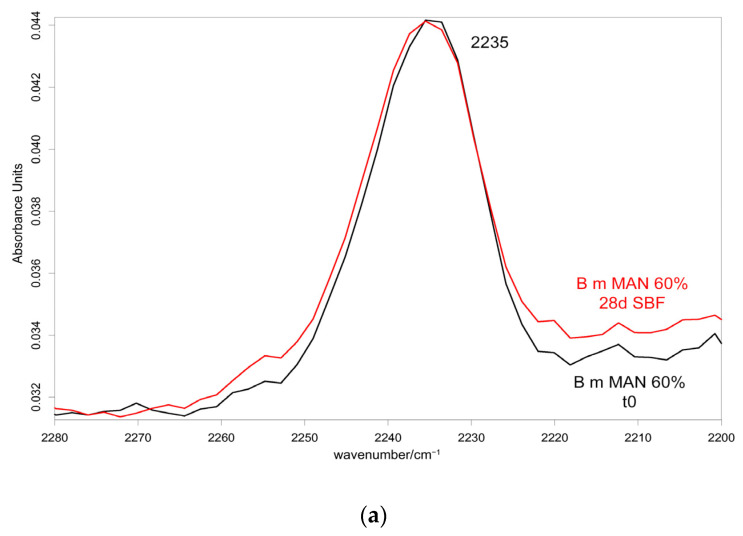
IR spectra recorded in the perpendicular orientation on the 60% MAN-grafted fibers before and after ageing for 28 days in SBF in the 2280–2200 cm^−1^ (**a**) and 1720–900 cm^−1^ (**b**) spectral ranges.

**Figure 16 molecules-28-02551-f016:**
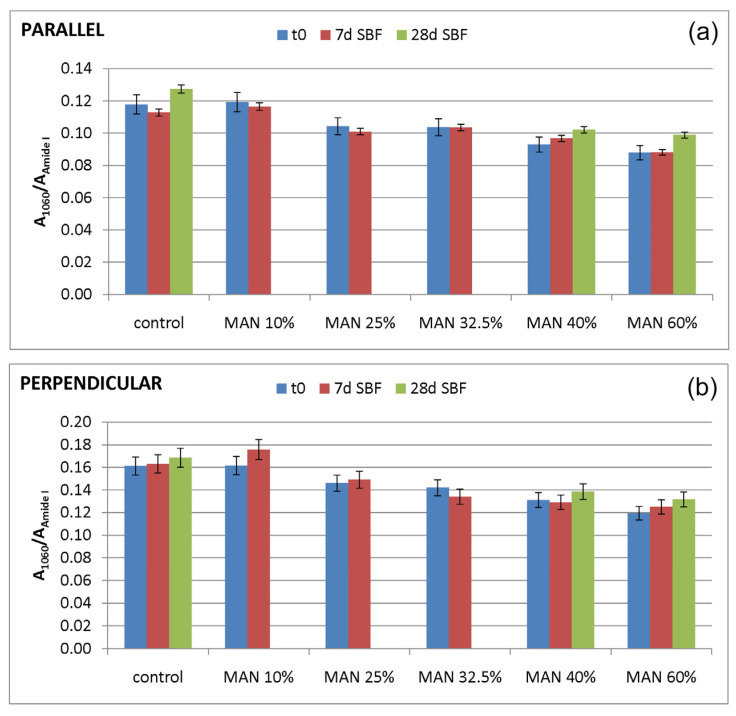
Values of the A_1060_/A_Amide I_ absorbance ratio as obtained from the IR spectra recorded in the parallel (**a**) and perpendicular (**b**) orientations on the samples under study (control = untreated fibers, and MAN-grafted fibers containing different amounts of MAN) before (t0) and after ageing of the fibers in SBF.

**Table 1 molecules-28-02551-t001:** MAN content *w*/*w*% as calculated from the linear regression of Raman I_2237_/I_Amide I_ and I_703_/I_Amide I_ of the grafted silk fibers under study before and after immersion in SBF for 7 and 28 days. The data are expressed as the average of three measures ± their standard deviation.

Sample	MAN Content *w*/*w*% from I_2237_/I_Amide I_			MAN Content *w*/*w*% from I_703_/I_Amide I_		
	t = 0	t = 7 days	t = 28 days	t = 0	t = 7 days	t = 28 days
MAN 10%	10.3 ± 0.9	8.1 ± 0.7		8.9 ± 1.2	7.8 ± 0.3	
MAN 25%	25.6 ± 2.4	23.7 ± 0.9		24.7 ± 2.5	24.0 ± 1.4	
MAN 32.5%	32.6 ± 1.0	28.4 ± 0.9		30.4 ± 1.0	28.9 ± 1.2	
MAN 40%	39.4 ± 4.1	39.1 ± 1.3	34.2 ± 2.4	39.5 ± 0.6	38.4 ± 3.6	33.8 ± 1.1
MAN 60%	60.1 ± 2.3	57.3 ± 0.8	52.6 ± 1.9	61.8 ± 2.4	60.4 ± 2.2	54.3 ± 5.6

## Data Availability

The data presented in this study are openly available as Appendix A.

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
