# Peer review of "Vibrational Study on the Structure, Bioactivity, and Silver Adsorption of Silk Fibroin Fibers Grafted with Methacrylonitrile"

_molecules, 2023, doi:10.3390/molecules28062551_

Round 1

Reviewer 1 Report

I see that the research work although it is narrow, but the way of presentation is very well. Starting from the selection of the title and experimentation, results discussion and correlation to the practical application, the authors didn't miss the track. They very well sticked to their point. I can comment on inclusion of some external studies but it will distract whole theme of the work. From my understanding of the work, the authors understand what they have to present in this article, they totally sticked to that one. Also, they very well described their results and therefore, I recommend this work for final publication in the present format.

Author Response

We thank the Reviewer for appreciating our work.

Reviewer 2 Report

The work entitled “Vibrational study on structure, bioactivity, and silver adsorp-2 tion of silk fibroins fibers grafted with methacrylonitrile” submitted for publication in MOLECULES by Michele Di Foggia, Masuhiro Tsukada and Paola Taddei is a robut spectroscopical work, and instead of the typical use of spectroscopy as a complement, here it is the main actor.

Statistically I am not convinced of the linear trend in Figure 9, and the %weight gain … the last value it not clear if it really leads to a linear trend or exponential. Could the authors general a value for 50 and 70? And the correlation in Figure 12, in blue with just 3 values is suspicious. Here if it is not possible to add values, it would be helpful to add a comment that the correlation us not statistically significant.

In the references us the right journal abbreviation when missing. Take for instance in Reference 25.

Overall, I consider that the manuscript deserves acceptance once the minor points are addressed

Author Response

  1. Figure 9 reports the %weight gain after immersion in Ag+-containing solutions as a function of %MAN content. We observed a linear correlation between the MAN-containing samples (which are reported as blue diamonds in the figure): the statistical parameters relative to the linear regression of data were R2 = 0.9474 and χ2 = 0.4481, this last corresponding to p > 0.9 that MAN content % and weight gain in the Ag+-containing solution are linearly dependants. The reference B. mori silk fibroin sample showed a %weight gain which was almost comparable with the highest MAN-containing samples; therefore, this last sample was not considered in the linear regression (for this reason, it is marked as a red square in the figure).

Figure 12 reports the %weight gain after immersion in SBF-containing solutions as a function of %MAN content at 7 and 28 days. The statistical parameters relative to the linear regression of the data relative to samples immersed in the bioactive solution up to 7 days support a linear correlation between the two parameters (R2 = 0.8738; χ2 = 0.8916 p > 0.9),  while data at 28 days were worse compared to the previous one (R2 = 0.7857, χ2 = 6.757), because of the reduced number of samples. Therefore, following the Reviewer’s suggestion, we inserted a comment about this finding.

  1. We revised all references and corrected mistakes.

Reviewer 3 Report

The article (Manuscript Number molecules-2254787) describes the crucial author's advances concerning the green method of preparation of silk fibroin fibers grafted with methacrylonitrile. As the authors stated, this solution has several advantages, because it is a green, easy, and low-cost approach. This work reduces the processing steps greatly. In my opinion, these findings make it sustainable and greener for the preparation of high-value-added products, such as in the field of direct biomedical applications, especially in bone tissue engineering and their interaction with Ag+ ions with enhanced antimicrobial properties, especially when we consider the 12 principles of green chemistry.
The article is very interesting, innovative, and written in a style that allows you to easily and quickly read what the author (authors) wanted to convey, but English should be improved in certain parts. For example, the abstract was written in such a way that I completely do not understand what the authors meant, and what was the purpose and effect of their research. The overall originality of the concept used here is high. Furthermore, it is based on a solid body of experimental work and is generally correct. I would recommend the publication of this paper in Molecules on the condition a major revision of the manuscript will be carried out and the following points will be taken into consideration.

Additional comments:
1.    The abstract should be rewritten - see details above.
2.    The Introduction section appears to be a collection of data from research papers, however, the author's self-opinion is of importance when drafting a section of this type.
3.    I think the authors need to make a bit less overclaiming approach and rather show other possibilities, except traditional hydrometallurgy processing.
4.    More detailed results discussion should be provided. The Results and discussion section appears to be a collection of data, however, the author's self-opinion is of importance when drafting a section of this type.
5.    Graphical abstract is not presented, the authors should put some effort to diversify it, and thus interest the potential reader.
6.    In this form, the article is very difficult to read. There is definitely no SI file in which all the procedures, description of the reagents used and some of the graphs should be included.
7.    In addition, all charts leave much to be desired. They are hard to read and prepared in a program that does not guarantee transparency.

After completing the above-mentioned corrections this work will be more readable. Therefore, it will be useful for the readers of Molecules.

Author Response

1.The abstract was rewritten to focus on the study’s purposes better.

  1. The authors structured the introduction to hierarchically guide the readers from a general overview of the application of natural fibers into different technological applications focusing on the application of modified silk fibroins in biomedicine. This structure partially reflects that the studied materials were first conceived for application in the textile sector (i.e., the early works of Prof. Tsukada, mentioned as references 14 and 15), but then, for their interesting physical and chemical properties, explored for possible application as biomaterials. Among the plethora of grafting or crosslinking agents for silk fibroins, we gain insights into acrylates and methacrylonitrile to give the reader an overview of the previous applications of this class of materials as biomaterials. Moreover, we used the introduction to stress the usefulness of spectroscopic techniques in studying silk-based materials. To better focus the authors’ self-opinion and the aim of the study, the parts concerning the results and the vibrational spectroscopic technique were shortened. With the same purposes, new references have been added, while others were eliminated [old 9,17,18].
  2. We eliminated the last sentence of the introduction, which could sound overclaiming, and inserted additional references about biomedical applications of silk fibers or membranes containing metal particles.
  3. Some parts were rewritten/added to make clearer the authors’ self-opinion, which was reported also in the Conclusions section.
  4.  The Graphical abstract was already present in the original submission: we hope it could attract readers’ interest as suggested by the Reviewer.
  5. An extensive supplementary material document was provided along with the main article, containing 2 tables and 29 figures. All the procedures and materials used in the study are reported in the “Materials and Methods” section of the Article.
  6. Following the Reviewer’s suggestion, we uploaded research data (mainly spectra and worksheets used to elaborate spectral data) as “Raw data” in Supplementary Material.

Round 2

Reviewer 3 Report

The article (Manuscript Number molecules-2254787) describes the crucial author's advances concerning the green method of preparation of silk fibroin fibers grafted with methacrylonitrile. As the authors stated, this solution has several advantages, because it is a green, easy, and low-cost approach. This work reduces the processing steps greatly. In my opinion, these findings make it sustainable and greener for the preparation of high-value-added products, such as in the field of direct biomedical applications, especially in bone tissue engineering and their interaction with Ag+ ions with enhanced antimicrobial properties, especially when we consider the 12 principles of green chemistry. All suggested changes were made (or discussed/clarified) by the authors. The results are informative, and the discussion is clear. The article is very interesting, innovative, and written in a style that allows you to easily and quickly read what the author (authors) wanted to convey. The overall originality of the described concept used here is adequate. To summarize, I think that this paper can be published as-is, it will be useful for readers of Molecules.